# Revisiting Positional Information in Transformers in the era of Fused Attention

## Abstract

Imparting positional information has been a crucial component in Transformers due to attention's invariance to permutation. Methods that bias attention weights, like Relative Positional Bias (RPB), have been preferred choice in more recent transformer-based architectures for vision. In parallel, fused attention has become the standard implementation for attention, largely thanks to open source solutions such as Flash Attention and FMHA. However, it is not trivial to fuse explicit biasing or masking of attention weights into a fused attention kernel without affecting its performance. In this scenario, position embeddings present themselves as a viable replacement for attention weight biases. Position embeddings are applied to the tokens directly, decoupled from the attention mechanism, thereby sidestepping the problems that arise with attention weight biases in fused kernels. In this work, inspired by the booming LLM landscape, we analyze the applicability of Rotary Position Embeddings (RoPE) as a replacement for RPBs in vision models. Unlike RPB which explicitly biases attention weights, RoPE biases the dot product inputs (query and key) directly and ahead of the attention operation. We empirically show the prowess of RoPE over RPBs in terms of accuracy and speed. We study multiple implementations of RoPE and show that it is sufficient to use only a fraction of hidden dimensions for RoPE to achieve competitive performance. We also develop a fast implementation for Axial RoPE. Together with the most performant fused attention implementations, and our fast RoPE implementation, we observe inference speedups compared to RPB with improved or similar accuracy. We foresee RoPE as a replacement for RPBs, paving the way for the widespread adoption of fused attention in transformer-based vision models.

## 1 Introduction

Self attention and transformers Vaswani et al. (2017) have proven to be powerful tools for learning from large amounts of unstructured data. The inception of Vision Transformers Dosovitskiy et al. (2021), or ViTs, further propelled the use of transformers for image and video modalities. ViT follows the isotropic architecture design of the Transformer, with a single downsampling step and identically shaped encoder layers. On the other hand, hierarchical vision transformers started to incorporate the CNN-like design Liu et al. (2021); Hassani et al. (2023); Hassani & Shi (2022); Ryali et al. (2023); Fan et al. (2021); Li et al. (2022), downsampling the token space gradually and increasing the number of attention heads. They also typically restrict their earlier attention layers to local or sparse patterns in order to avoid scaling issues resulting from performing global self attention.

The widespread usage of attention in language and vision inspired the creation of fused attention implementations like Flash Attention Dao et al. (2022); Dao (2023) and FMHA Lefaudeux et al. (2022). These implementations are functionally equivalent to a BMM-style implementation in a deep learning framework like PyTorch Paszke et al. (2019), but provide significant improvements in performance and activation memory footprint by keeping attention weights in fast local memory and fusing the second half of the operation, instead of storing attention weights as an additional activation to the relatively slower global memory, thereby reducing the number of expensive global memory accesses.

In all transformers, tokens are the smallest unit of representation. Since the attention mechanism is invariant to the permutation of these tokens, additional positional biases are added to inject spatial

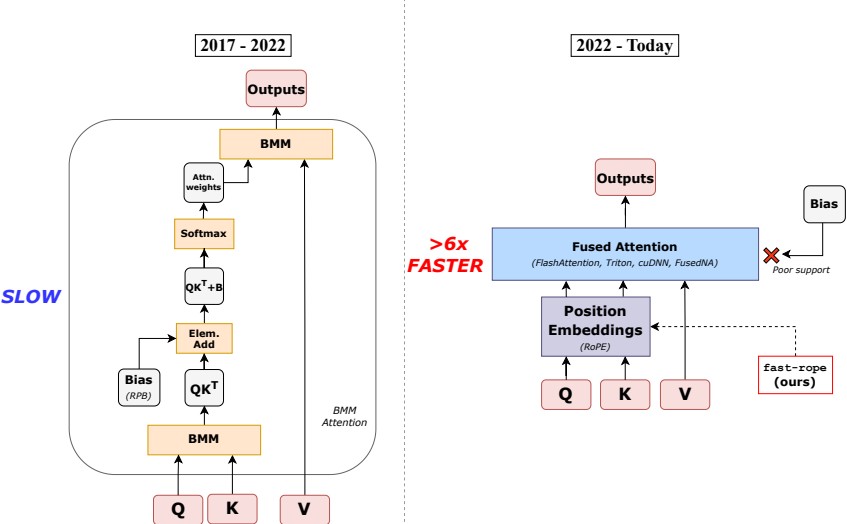

Figure 1: **Attention weight biases in BMM-style and fused attention**: Attention weight biases (like RPB) are added directly to the attention map. This bottlenecks the backward pass in fused attention kernels since the update for bias tensor is a reduction operation. Due to this, many implementations of fused attention implementations do not support explicit biases or attention masks. Using position embeddings like RoPE will enable less restricted usage of fused attention implementations. We note that xFormers' FMHA and some others support explicit attention weights and masking, but they rarely succeed in hiding the additional and sometimes considerable latency from the bias.

information into the transformer. The original ViT used Absolute Position Embeddings (APE) to solve this problem. In the years that followed, encoding positional information in the form of attention weight biases became a popular choice in transformer-based models in vision, among which, Relative Positional Bias (RPB) Shaw et al. (2018) has been one of the most popular. However, RPB, and attention weight biases in general, can somewhat greatly hinder the performance of fused attention implementations. While they are a barely noticeable elementwise operation in forward pass, the backward pass for attention weight biases is a reduction operation. This makes it non-trivial to fuse the already complex fused attention backward kernel together with that of position biases. Additionally, incorporating such biases in newer implementations of fused attention requires an unjustifiably significant engineering effort. The recently released Flash Attention V2 and V3 Dao (2023); Shah et al. (2024) never supported explicit masking or biasing. To date, very few implementations, namely the xFormers' FMHA, offer such features. In addition to the effort required to implement, hiding the latency of the softmax operation in pipelined attention kernels such as FAv3 Shah et al. (2024) is already very challenging, and supporting explicit attention weight biasing or masking will add to that latency and easily expose it. We illustrate this in Figure 1.

On the other hand, position embeddings are usually decoupled from the attention mechanism and are applied to the input tokens instead, ahead of the attention operation. Originally these embeddings were applied to the tokens only once at the very beginning of the model. This approach is commonly referred to as Absolute Positional Embedding (APE). However, Rotary Position Embeddings (RoPE) Su et al. (2021) have become the de-facto choice in large language models Touvron et al. (2023a;b); Chiang et al. (2023), and are slowly making their way into vision models as well Crowson et al. (2024); Karras et al. (2022). We compare these three methods for introducing positional information (RPB, APE, and RoPE) in Figure 2. Compared to APE, RoPE can be seen as much more flexible generalization. Compared to RPB, the advantages of RoPE are threefold: 1. RoPE is a static position embedding mechanism; RoPE can be interpolated for varying input resolutions without retraining or finetuning. 2. the forward and backward pass of RoPE are both element-wise operations, for which developing highly parallelized SIMT implementations and kernel fusions are much easier. Lastly, since RoPE is agnostic to the dot-product attention operation, one can use the best available fused attention implementation for their use case, and to its full potential.

Inspired by this, we thoroughly study the applicability of RoPE in transformer-based models for vision. Our main contributions are as follows:

1. We present the scaling and implementation-related challenges in using RPB, or any explicit attention bias, in the context of fused attention. Positional biases, while very simple elementwise operations in their forward pass, are a reduction in their backward pass, making their fusion into complex fused attention kernels very challenging.

2. We empirically show the improvement from using Rotary Position Embeddings over RPB. We show consistent improvements across three model families: ViT, Swin, and NAT, with varying model sizes (19 million to 89 million parameters). These models also cover three different attention patterns: self attention, windowed attention and neighborhood attention. We achieve noteworthy gains across all models.

3. We develop an efficient CUDA implementation for RoPE with an easy-to-use Python wrapper. We carefully benchmark it and show its speedup against using RPB.

4. We carefully study multiple implementations of Rotary Position Embeddings and present an analysis of using RoPE in transformer-based vision models. We empirically show that one can use only a fraction of hidden dimensions for RoPE and still achieve competitive performance. We introduce a hyperparameter $k_{rope}$ and analyze its effect on multi-resolution performance.

## 2 RELATED WORK

In this section, we review some prominent transformer-based architectures for vision, as well as current methods for introducing spatial biases, and the effect of using positional biases with fused attention implementations.

### 2.1 VISION TRANSFORMERS AND HIERARCHICAL VISION TRANSFORMERS

After the inception of the original Vision Transformer (ViT)Dosovitskiy et al. (2021), a considerable research effort has been towards understanding Ghiasi et al. (2022); Raghu et al. (2021) and improving ViTs. Most notably, many works are inspired by the efficient design of CNNs and have transformer the isotropic ViT into a multi-level hierarchical vision transformer Liu et al. (2021); Fan et al. (2021); Li et al. (2022). The networks reduce the spatial dimensions of the feature map at every level with increasing channels (attention heads). Similar to CNNs, many found that tokens in the earlier layers and levels of these models attend more locally, and those in later layers and levels attend more globally Raghu et al. (2021). This has propelled the development of hierarchical vision transformers with restricted local attention Hassani et al. (2023); Hassani & Shi (2022); Ryali et al. (2023).

### 2.2 ADDING POSITIONAL BIASES TO VISION TRANSFORMERS

Transformers are primarily comprised of linear layers and attention, both of which are invariant to token permutation, which naturally led to researchers introducing positional information into their models. After the inception of the original transformer architecture Vaswani et al. (2017), many new methods were introduced to add position information to transformers Ke et al. (2020); Huang et al. (2020). ViT Dosovitskiy et al. (2021) used absolute sinusoidal position embeddings used in the original Transformer Vaswani et al. (2017). Relative Positional Biases (RPBs) Shaw et al. (2018) quickly became the de facto method used in a plethora of hierarchical vision transformers. More recently, Rotary Position Embeddings Su et al. (2021) became the norm in billion-parameter models like LLaMA Touvron et al. (2023a) and its derivatives Touvron et al. (2023b); Chiang et al. (2023). RoPE enjoys several benefits, like usability in long contexts, better training stability, and decaying influence with increasing relative distance, to name a few. This makes RoPE a more scalable alternative to RPBs. However, RoPE has not yet been as widely adopted in vision models, and we aim to shed light in this direction through this work. We find that 2D RoPE Heo et al. (2024); Jeevan & Sethi (2022) and AS2DRoPE Chu et al. (2024) are suboptimal extensions of original RoPE in the context of vision transformers. We illustrate the difference between APE, RPB and RoPE in Figure 2. We observe a clear trend; newer and larger models often prefer position embeddings over attention weight biases.

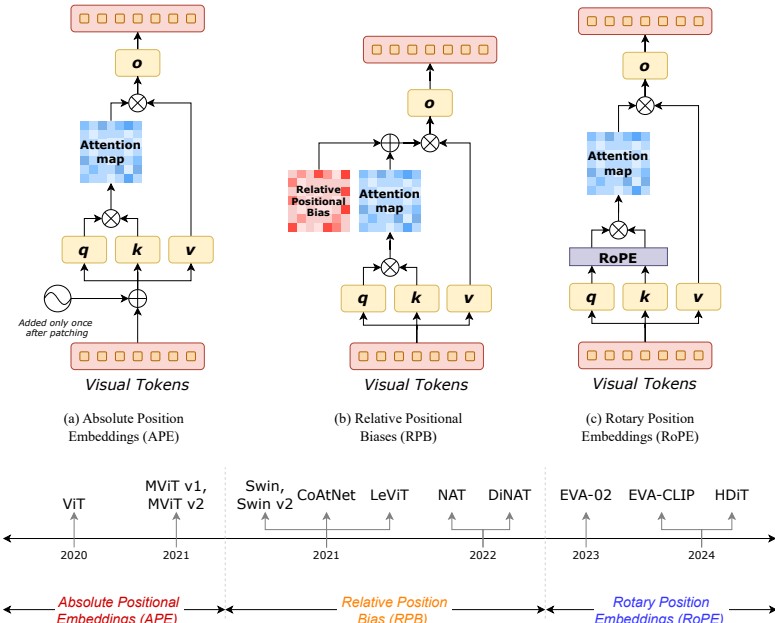

Figure 2: **Comparison between types of position embeddings.** Absolute Position Embeddings are applied once to the tokens after patchifying, but RPB and RoPE are both applied in every attention block. RPBs are added to the attention map itself, whereas RoPE is applied to queries and keys. We observe that prominent models used APE and RPB in the early years of vision transformers. However, newer models like HDiT and EVA-CLIP, which have been scaled to up to 18 billion parameters, opt for the more scalable Rotary Position Embeddings. We see a growing trend towards the applying position embeddings to the tokens themselves in contrast to biasing attention weights.

## 2.3 FUSED ATTENTION IN VISION TRANSFORMERS

For most of its history, dot-product attention, one of the primary operations in the Transformer, has been implemented as back-to-back batched matrix multiplications (BMMs), now commonly referred to as BMM-style attention. The first BMM computes the dot products between query and key tokens, the softmax of which produces attention weights. Attention weights are then "applied" to the values by taking their weighted average using the corresponding scores in the weight matrix through the second BMM. At scale, this implementation can quickly become bounded by memory bandwidth and capacity. In the case of self attention, in addition to a quadratic time complexity, the memory footprint is also quadratic. This inspired "fused" attention implementations, the first practical example of which is Flash Attention Dao et al. (2022), which was later followed by Flash Attention V2 Dao (2023), FMHA Lefaudeux et al. (2022), and many more implementations. These methods successfully fuse the two BMMs and the softmax into one kernel by using partial softmax aggregation (since softmax involves a reduction), allowing them to scale to large sequence lengths. Through doing so, they improve performance by significantly reducing accesses to global memory, and instead keeping attention weights on much higher throughput local memory (shared memory). In addition, the global memory footprint is also reduced significantly. As a consequence, fused implementations are naturally less flexible in terms of allowing manipulation of attention weights. This presents a challenge to positional biases, which, even when implemented, can noticeably impact the performance of fused attention kernels.

In light of this, Rotary Position Embeddings are much better suited for fused attention because they do not operate on attention weights directly. Instead, RoPE is applied to to the query and key tensors prior to attention. This decoupling of positional biases and attention computation can significantly improve model performance. This work exploits this fact further by developing a fast RoPE implementation suited specifically for vision models.

## 3 METHOD

In this section, we will outline the usage of position embeddings in vision transformers. First, we will go over attention weight biases in Section 3.1 and then go on to formalize RoPE in Section 3.2. We comprehensively explain the existing 2D variants of RoPE in Section 3.3. Lastly, in Section 3.4, we will discuss some practical implications of using RoPE and present our fused implementation.

### 3.1 ATTENTION WEIGHT BIASES

Attention weight biases have become a common choice to add spatial biases in vision transformers. Attention weight biases assign a bias value for every query-key pair in the attention map. Techniques like Relative Positional Biases (RPB) use the relative position of query and key tokens to add a specific bias term to them. Attention weight bias is added directly to the raw attention weights calculated by taking the dot product between queries and keys. Formally, in a feature map of the size $(H, W)$ we will have queries $Q \in \mathbb{R}^{HW \times d}$ and keys $K \in \mathbb{R}^{HW \times d}$ where $d$ is the channel dimension. A bias $B \in \mathbb{R}^{HW \times HW}$ will be added to the attention weights as follows:

$$A = (QK^T) + B \tag{1}$$

In the case of RPB, the bias $B$ is parameterized as a smaller tensor but "viewed" as a tensor with the same shape as the attention weights. Generally, attention weight biases in vision transformers are learnable and thus need to be interpolated if the spatial resolution of the input image changes. Moreover, they cause a bottleneck in the backward pass of any fused attention kernel. We delve into practical implications of attention weight biases in Section 3.4.

### 3.2 ROTARY POSITION EMBEDDINGS

Rotary Position Embeddings (RoPE) Su et al. (2021) were proposed to equip tokens in language models with stronger positional information. RoPE applies position embeddings based on the global position of the token in the sequence, but the actual embedding function is derived to keep the relative distances amongst two tokens intact irrespective of their global positions.

Rotary Position Embeddings impart spatial bias by chunking the feature vector of dimension $d$ into $d/2$ chunks of two elements each, and rotating each chunk in the Argand plane. The angle of rotation is decided based on the token's position in the sequence. Formally, considering a token $\mathbf{x}$ at index $t$ in a sequence of length $N$, its resulting embedding $\mathbf{x}^t$ will be given by

$$\begin{aligned} \mathbf{x}_{j,k}^t &= \mathbf{x}_{j,k} e^{i\theta_j^t} \ \ \forall j \\ \theta_j^t &= t * f_\theta(j, d) \end{aligned} \tag{2}$$

where $j, k$ are the dimension indices and $j \in \{0, ..., d/2 - 1\}$ and $k = j + d/2$. Here $f_\theta(j, d)$ is the rotation angle generator – it produces an angle for each dimension index, given $j$ and $d$. Conventionally, the angle generator is given by $f_\theta(j, d) = 10000^{-2j/d}$ in LLMs like Llama Touvron et al. (2023a). Now, consider a query and key $\mathbf{q}^m, \mathbf{k}^n$ at the positions $m$ and $n$ respectively with RoPE applied according to their positions. Their corresponding entry $A_{m,n}$ in the attention matrix will be given by

$$\begin{aligned} A_{m,n} &= \mathbf{q}^m \cdot \mathbf{k}^n \\ A_{m,n} &= \sum_j \mathbf{q}_{j,k} \mathbf{k'}_{j,k} \ e^{i(\theta_j^m - \theta_j^n)} \end{aligned} \tag{3}$$

where $\mathbf{q}_{j,k}$ is the unmodified $j, k$ chunk of the query vector and $\mathbf{k'}_{j,k}$ is the unmodified complex conjugate of the $j, k$ chunk of the key vector. "·" represents dot product of two vectors.

We make some key observations from Equation 3. First, the relativity of two tokens is captured by each chunk in the exponent $\theta_j^m - \theta_j^n$. Second, as $m - n$ increases, $|\sum_j e^{i(\theta_j^m - \theta_j^n)}|$ decreases, implying decay of influence when the relative distance between the tokens increases. This makes RoPE a preferred choice for injecting positional bias in both self and local attention mechanisms, as the property of relativity holds in both cases. We discuss practical benefits of RoPE in Section 3.4.

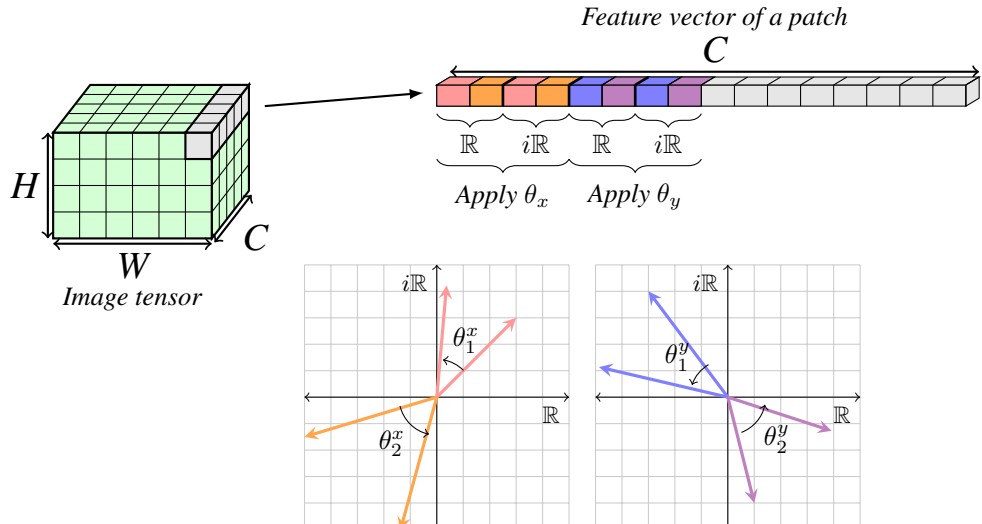

*Rotation of chunks w.r.t. positions in X and Y axis in the image.*

Figure 3: **Illustration of RoPE for images.** In this figure, two elements of the same color denote a single point on the Argand plane. We interleave the elements to denote that half of the elements constitute the real part and the other half constitute the complex part of the points. Here $k_{rope} = 2$, thus, we use only the first half of the feature to encode positional information.

### 3.3 AXIAL ROTARY POSITION EMBEDDINGS

We introduced RoPE for a one-dimensional sequence of tokens in Section 3.2. We will now extend it for images, which have a two-dimensional array of tokens. We will elucidate the intricate changes made to RoPE to make it suitable for multi-resolution 2D settings.

#### 3.3.1 EXTENDING ROPE FOR 2-D TOKEN ARRAYS

Consider a 2-D feature map of the spatial dimensions $(H, W)$. To extend the current approach of RoPE from the 1-D setting, we will simply decompose the two-dimensional position index into two one-dimensional positional indices. Naturally, this means that each position will have an index corresponding to the X-axis and one corresponding to the Y-axis. We will then assign a rotation angle for each position index in both axes.

Secondly, we will divide the number of dimensions in two parts – one part for one axis each. We will then proceed to further divide each part into two to represent the real and complex parts each, creating a total of four parts of the entire feature vector. The first and third chunks are treated as the real and complex parts for the position in Y-axis, and the second and fourth chunks are treated as the real and complex parts for the token's position in X-axis. This is illustrated in Figure 3. Formally, this can be expressed as follows. Given an unmodified query vector $\mathbf{q}$ at the position $(y, x)$, its resulting embedding $\mathbf{q}^{y,x}$ will be given by

$$
\begin{aligned}
\mathbf{q}^{y,x} &= \begin{cases} q_{j,k} \, e^{i\theta_j^x} \ \ \forall \, j \in \{0, ..., d/4\} \\ q_{j,k} \, e^{i\theta_j^y} \ \ \forall \, j \in \{d/4, ..., d/2\} \end{cases} \\
\theta_j^x &= x * f_\theta(j, d) \\
\theta_j^y &= y * f_\theta(j - d/4, d)
\end{aligned}
\tag{4}
$$

and $k = j + d/2$. An important thing to note is the angle generator is the same for both axes, however, we subtract the dimension index by $d/4$ so that both axes are treated in identical manner. In other words, the first $d/4$ indices hold the positional information in Y-axis and the second $d/4$ indices do the same for X-axis. The third and fourth $d/4$ chunks follow the same ordering, and carry the

| | 2D RoPE | Axial RoPE |
|---|---|---|
| *Position co-ordinates* | Absolute indices | Interpolated between (-1, 1) |
| *Angles of rotation* | $100^{-4j/d}$ | $\exp\left(j*(log(10\pi) - log(\pi))/d\right)$ |
| *Default $k_{rope}$* | 1 | 2 |
| *Shared $\theta$ for all heads* | Yes | No |

Table 1: **Comparing the design decisions in 2D RoPE and Axial RoPE.** We note these design decisions are inspired by different applications – 2D RoPE has been inspired by works in the LLM community whereas Axial RoPE has been inspired by works in the image generation space.

complex parts of the first and second chunks respectively. This way, we can encode a 2-D position in a single feature vector. Additionally, this approach can be extended to an arbitrary number of spatial dimensions, $n$, with the only constraint $d \pmod{2n} \equiv 0$. Since this implementation expands on 1D RoPE for the 2D case, we henceforth refer to this as *2D RoPE*. Scalable vision models use this flavor of 2D RoPE, and is directly inspired from LLMs like LLaMA.

### 3.3.2 IMPROVING 2D ROTARY POSITION EMBEDDINGS

Image modality is vastly different from language requires a separate treatment compared to language. Modern generative models Karras et al. (2022); Nawrot et al. (2021); Crowson et al. (2024) use a specific variant of 2D RoPE implementation which makes it more suitable for the vision domain. The following changes are made to 2D RoPE, and we call the resulting variant as *Axial RoPE*.

**Sampling $y, x$ between -1 and 1:** Conventionally, the token index is the multiplicative factor that changes the rotation angle to signify different positions in the sequence. Language is inherently causal, and thus this approach makes sense in the language domain. However, an image does not follow the same notion. Thus, we sample the multiplicative factor by linearly interpolating between -1 and 1 for the spatial dimensions. We later empirically show that this is indeed equivalent than using the plain token indices as the multiplicative factor.

**Bounded log sampling of rotation angles:** The original RoPE implementation generates rotation angles through the function $f_\theta(j, d) = 10000^{-2j/d}$ ($j$ is the dimension index). We note that using an bounded log-sampling for the base rotation angle causes the index-specific rotation angle to fluctuate gracefully. We log-sample our angle between $\pi$ and $10\pi$. Our angle generator function is given by

$$f_\theta(j, d) = \exp\left(j*(log(10\pi) - log(\pi))/d\right) \tag{5}$$

**Applying RoPE to only a fraction of the hidden dimensions:** We empirically observe that applying RoPE to a fraction of hidden dimensions retains, or in some cases exceeds the performance of applying RoPE to the full feature vector. We hypothesize that this is because applying RoPE to all dimensions greatly mutilates the actual semantic information in the tokens. Additionally, applying RoPE to a fraction of hidden dimensions is sufficient to impart the necessary positional information. We divide the total dimensions by a positive integer divisor and apply RoPE to the resultant number of dimensions. We will henceforth refer to this divisor as $k_{rope}$. In the following section, we will present empirical results analyzing $k_{rope} \in \{1, 2, 4, 8, 16\}$. We will also draw correlations between multi-resolution performance and the fraction of hidden dimensions used.

**Shared angles of rotation for all heads:** One of the differences between 2D RoPe and Axial RoPE is the usage of same rotation angles for all heads. Axial RoPE uses different angles for all heads, and on the other hand, 2D RoPE uses the same rotation angles for all heads. We empirically study the effects of using shared or non-shared angles of rotation for 2D and Axial RoPE. Note that for non-shared rotation angles, RoPE is applied to the first $d/(heads * k_{rope})$ dimensions for each head.

We summarize the differences between 2D and Axial RoPE in Table 1. In order to use RoPE to its best, we encourage practitioners to perform sweeps over $k_{rope}$ and using shared or non-shared angles with Axial and 2D RoPE for their specific use case. In this paper, we will empirically study the effect of these design choices on the ImageNet-1k dataset. We study the multi-resolution performance of each variant and draw connections with the implementation specifics and the choice of hyperparameters.

### 3.4 Hardware efficiency and scalability of RoPE vs RPB

As mentioned, Rotary Position Embeddings are applied to the query and key tensors, as opposed to attention weights in the case of RPBs. Moreover, RoPE is a more complex operation: for each of the two tensors, two elements are read from the feature vector, to which rotation is applied through reading one more element from the $\theta$ tensor, before the elements are stored back into the original tensor. This operation consists of multiple elementwise operations, all of which can grow quickly into a memory-bandwidth-bound bottleneck in eager mode. On the other hand, RPB is typically only a single elementwise operation, leading one to think that in theory RPB is more efficient.

However, there are two key issues with RPB in terms of performance: 1. though it is a single elementwise operation, we do not always have access to attention weights, a clear example of which is fused implementations. For example, Flash Attention V2 Dao et al. (2022) does not support attention biases at all, while FMHA Lefaudeux et al. (2022) only supports when they are fully materialized in global memory, and padded to meet memory alignment requirements, which can undo some of the performance improvements, as the positional biases will consume the same amount of memory that attention weights would. 2. Loading fully materialized attention biases or masks in fused attention kernels burdens their scalability. Fused attention kernels are typically compute bound, whereas unfused implementations of attention are heavily bound by memory bandwidth. Adding $O(n^2)$ loads into fused attention kernels can make them memory bandwidth bound again at scale. FNA Hassani et al. (2024) on the other hand only supports RPB in the forward pass. 3. RPB's backward pass is typically not an elementwise operation. Local / sparse attention will turn the RPB gradient into a **reduction** operation, which is considerably more difficult to performance optimize compared to elementwise operations. While fusing an elementwise operation into complex fused attention kernels such as Flash Attention V2 Dao (2023) is relatively trivial, fusing reduction operations into any kernel, especially fused attention kernels, is typically non-trivial. RoPE however is an elementwise operation both in the forward and in the backward pass. In fact, the only difference between the forward and backward pass is the sign of one element, which means:

- RoPE can be applied efficiently in both the forward pass and backward pass,
- RoPE's forward and backward pass implementations are almost identical,
- RoPE is completely agnostic to the attention operator, making it compatible with all implementations out of the box.

Having said that, a vanilla PyTorch implementation made RoPE *slower* than RPB, even when using powerful fused attention implementations. This is primarily because multiple elementwise operations used in RoPE are not automatically fused into a single one, and while using tools such as `torch.compile()` does exactly that, they simply do not improve performance enough to justify switching from RPB. This motivated us to develop `fast-rope`, a fused CUDA implementation performing Axial RoPE on feature tensors. The implementation follows HDiT's specifications Crowson et al. (2024): it can read operands of mixed precision levels, but computation is done strictly in higher precision. This allows us to perform the operation in place on the original tensor, without any extra type cast operations or kernel calls. We plan to open-source our library and release a Python package which will make using our library as easy as performing one function call.

## 4 Experiments

We demonstrate the capabilities of Axial RoPE on four model families – ViT Dosovitskiy et al. (2021), Swin Liu et al. (2021), NAT Hassani et al. (2023) and DiNAT Hassani & Shi (2022) on the ImageNet-1k dataset Russakovsky et al. (2015). Furthermore, we present the throughput gains obtained by using fused implementation of Axial RoPE combined with fused attention implementations. Lastly, we present our analysis on different existing RoPE methods, and discuss why we picked Axial RoPE.

### 4.1 Classification on ImageNet-1k

We evaluate Axial RoPE on the ImageNet-1k dataset and report the validation set accuracy in Table 2. Specifically, we report the scores for three cases – without using any positional information, with RPB, and with Axial RoPE ($k_{rope} = 2$). We also report acheived throughput for all these cases. We

| Attention mechanism | Model | | RPB (%) | No bias (%) | Axial RoPE (%) |
|---|---|---|---|---|---|
| *Neighborhood Attention* | **NAT** | *Mini* | 81.8 | 81.3 (-0.5) | 82.1 (+0.3) |
| | | *Tiny* | 83.1 | 82.5 (-0.6) | 83.2 (+0.1) |
| | | *Small* | 83.6 | 83.3 (-0.3) | 83.8 (+0.2) |
| | | *Base* | 84.3 | 84.0 (-0.3) | 84.5 (+0.2) |
| | **DiNAT** | *Mini* | 81.7 | 81.5 (-0.2) | 81.9 (+0.2) |
| | | *Tiny* | 82.7 | 82.6 (-0.1) | 83.0 (+0.3) |
| | | *Small* | 83.8 | 83.6 (-0.2) | 83.9 (+0.2) |
| | | *Base* | 84.4 | 84.1 (-0.3) | 84.5 (+0.1) |
| *Window Attention* | **Swin** | *Tiny* | 81.2 | 80.2 (-1.0) | 81.5 (+0.3) |
| | | *Small* | 83.0 | 81.9 (-1.1) | 83.1 (+0.1) |
| | | *Base* | 83.5 | 82.7 (-0.8) | 83.7 (+0.2) |
| *Self Attention* | **ViT** | *Small* | 81.2 | 79.0 (-2.2) | 81.4 (+0.2) |
| | | *Base* | 82.6 | 81.2 (-1.4) | 82.8 (+0.2) |

Table 2: **ImageNet-1k top-1 accuracies:** We present the top-1 accuracy on the ImageNet-1k validation split. We observe a accuracy over RPB across all model architectures and sizes.

| Model | | RPB | | No bias | | | Axial RoPE | | |
|---|---|---|---|---|---|---|---|---|---|
| | | *BMM-style* | *Fused* | *BMM-style* | *Fused* | | *BMM-style* | *Fused* | |
| **NAT** | *Mini* | 2664 | 3774 | 2871 (+7.8%) | 3870 (+2.5%) | | 2769 (+3.9%) | 3772 (-0.1%) | |
| | *Tiny* | 1948 | 2806 | 2095 (+7.5%) | 2898 (+3.3%) | | 2025 (+4.0%) | 2810 (+0.1%) | |
| | *Small* | 1335 | 1935 | 1436 (+7.6%) | 2000 (+3.4%) | | 1387 (+3.9%) | 1931 (-0.2%) | |
| | *Base* | 1029 | 1511 | 1113 (+8.2%) | 1564 (+3.5%) | | 1070 (+4.0%) | 1517 (+0.4%) | |
| **DiNAT** | *Mini* | 2558 | 3948 | 2802 (+9.5%) | 4053 (+2.7%) | | 2704 (+5.7%) | 3922 (-0.7%) | |
| | *Tiny* | 1857 | 2943 | 2043 (+10.0%) | 3028 (+2.9%) | | 1965 (+5.8%) | 2932 (-0.4%) | |
| | *Small* | 1342 | 2223 | 1485 (+10.7%) | 2292 (+3.1%) | | 1429 (+6.5%) | 2200 (-1.0%) | |
| | *Base* | 979 | 1592 | 1081 (+10.4%) | 1638 (+2.9%) | | 1041 (+6.3%) | 1587 (-0.3%) | |
| | | *BMM-style* | *FMHA* | *FAv2* | *BMM-style* | *FMHA* | *FAv2* | *BMM-style* | *FMHA* | *FAv2* |
| **Swin** | *Tiny* | 3018 | 3444 | - | 3114 (+3.2%) | 3535 (+2.6%) | 3472 | 2982 (-1.2%) | 3394 (-1.5%) | 3329 |
| | *Small* | 1902 | 2173 | - | 1966 (+3.4%) | 2237 (+2.9%) | 2196 | 1875 (-1.4%) | 2146 (-1.2%) | 2106 |
| | *Base* | 1450 | 1658 | - | 1501 (+3.5%) | 1705 (+2.8%) | 1676 | 1431 (-1.3%) | 1639 (-1.1%) | 1611 |
| **ViT** | *Small* | 4665 | 7864 | - | 5256 (+12.7%) | 8207 (+4.4%) | 8593 | 5144 (+10.3%) | 7964 (+1.3%) | 8337 |
| | *Base* | 2178 | 3343 | - | 2423 (+11.2%) | 3468 (+3.7%) | 3598 | 2367 (+8.7%) | 3387 (+1.3%) | 3519 |

Table 3: **Inference throughput:** We present the inference throughput for all models, using different attention mechanisms. Here "*BMM-style*" represents attention implemented with native PyTorch, without any optimizations. "*Fused*" represents Fused Neighborhood Attention (FNA) Hassani et al. (2024) in the case of NA-based models. "*FMHA*" represents xFormers's implementation Lefaudeux et al. (2022), and "*FAv2*" represents Flash Attention V2 Dao (2023). All numbers represent the throughput in images per second. We report the improvements over RPB in green.

observe consistent accuracy improvements across all model families spanning across a wide range of parameter counts and FLOPs. Interestingly, we see that larger models with high parameter counts, like the "*Base*" variants also enjoy the same performance boosts as the smaller models. We provide additional evaluations on ImageNet A Djolonga et al. (2021), ImageNet RHendrycks et al. (2021), ImageNet ReaLBeyer et al. (2020) and ImageNet V2 Recht et al. (2019) in Appendix F. We clearly observe that RoPE outperforms RPB on all models, irrespective of their size or attention mechanism.

## 4.2 BENCHMARKING FAST-ROPE

In Table 3, we present throughputs of models with Axial RoPE using our fused implementation against models with RPB. We report inference throughput (i.e. forward pass throughput) using both BMM-style and fused attention implementations [1]. For Neighborhood Attention, those would be the GEMM-based and fused kernels from NATTEN. For Swin and ViT, that would be xFormers' FMHA and Flash Attention V2. We use Automatic Mixed Precision (AMP) in all tests to do FP16 inference. We observe considerable gains in almost all models, while being roughly equal in the case of Swin.

---

[1]All performance measurements were benchmarked on the A100-SXM4.

| $k_{rope}$ | 128 px | | 192 px | | 224 px | | 256 px | | 320 px | | 384 px | | 480 px | | 512 px | |
|---|---|---|---|---|---|---|---|---|---|---|---|---|---|---|---|---|
| | 2D | Axial | 2D | Axial | 2D | Axial | 2D | Axial | 2D | Axial | 2D | Axial | 2D | Axial | 2D | Axial |
| RPB | 34.19 | | 78.62 | | 81.22 | | 81.21 | | 79.77 | | 77.97 | | 75.19 | | 73.9 | |
| 1 | **70.53** | 62.85 | 80.09 | 78.47 | 81.42 | 80.85 | 81.69 | 80.32 | 81.31 | 79.85 | 80.08 | 78.57 | 76.37 | 75.84 | 74.73 | 74.69 |
| 2 | 69.81 | 65.43 | 79.89 | 80.11 | 81.37 | 81.43 | 81.84 | 81.77 | 81.29 | 81.28 | 79.02 | 79.90 | 72.78 | **77.41** | 69.78 | **76.47** |
| 4 | 70.02 | 65.76 | **80.18** | 80.19 | 81.46 | 81.47 | **81.96** | 81.78 | 81.20 | 80.80 | 77.54 | 79.45 | 68.13 | 76.37 | 63.80 | 75.16 |
| 8 | 69.37 | 41.59 | 79.81 | 79.61 | 81.14 | **81.51** | 81.73 | 81.78 | **81.68** | 80.78 | **80.34** | 78.77 | 76.77 | 76.05 | 75.48 | 75.01 |
| 16 | 63.6 | 56.13 | 79.19 | 79.76 | 80.77 | 81.34 | 81.16 | 81.70 | 80.32 | 81.12 | 78.17 | 79.78 | 74.00 | 76.68 | 72.36 | 75.82 |

Table 4: **Multi-resolution performance for different values of $k_{rope}$ with shared angles for all heads.** We highlight the best performing entry in every resolution group.

| $k_{rope}$ | 128 px | | 192 px | | 224 px | | 256 px | | 320 px | | 384 px | | 480 px | | 512 px | |
|---|---|---|---|---|---|---|---|---|---|---|---|---|---|---|---|---|
| | 2D | Axial | 2D | Axial | 2D | Axial | 2D | Axial | 2D | Axial | 2D | Axial | 2D | Axial | 2D | Axial |
| RPB | 34.19 | | 78.62 | | 81.22 | | 81.21 | | 79.77 | | 77.97 | | 75.19 | | 73.9 | |
| 1 | 67.02 | 62.25 | 79.17 | 78.36 | 80.75 | 80.91 | 81.17 | 80.44 | 80.28 | 80.17 | 78.49 | 78.95 | 74.40 | 76.53 | 73.10 | 75.6 |
| 2 | **68.10** | 67.18 | 79.45 | 80.03 | 81.00 | 81.41 | 81.44 | **81.67** | 80.44 | **81.05** | 77.86 | **79.64** | 71.31 | **77.12** | 68.86 | **75.89** |
| 4 | 67.56 | 65.26 | 79.41 | 79.55 | 80.93 | 81.41 | 81.38 | 81.13 | 80.62 | 80.54 | 78.36 | 79.14 | 73.60 | 75.96 | 71.32 | 74.61 |
| 8 | 66.31 | 51.09 | 79.41 | 77.64 | 81.11 | **81.42** | 81.47 | 79.81 | 80.79 | 77.64 | 79.20 | 76.23 | 75.16 | 74.17 | 73.37 | 71.76 |
| 16 | 68.02 | 53.97 | 79.73 | 78.34 | 81.18 | 81.16 | 81.43 | 80.08 | 80.55 | 79.62 | 78.30 | 77.81 | 73.73 | 73.95 | 71.64 | 72.45 |

Table 5: **Multi-resolution performance for different values of $k_{rope}$ with non-shared angles for all heads.** We highlight the best performing entry in every resolution group.

We attribute the slowdown in Swin to the nature of $\theta$ tensor, as it is larger in Swin due to the presence of the batch dimension. Even with this disadvantage, RoPE is able to catch up to RPB's throughput. Additional benchmarks of `fast-rope` are included in Appendix A.

### 4.3 ANALYZING DESIGN DECISIONS IN ROPE

We will now analyze the differences between the two implementations mentioned above – 2D RoPE and Axial RoPE. We will consider the choices of $k_{rope}$ and whether all heads use the same rotation angles. Through this analysis, we aim to empirically study the effect of these hyperparameters on RoPE and its multi-resolution performance on a wide range of testing resolutions. We perform all our ablations on ViT-Small trained with 224 px resolution.

Table 4 presents the performance when all heads share the same rotation angles. We make a striking observation – 2D RoPE with a high value of $k_{rope}$ often performs best. In the case where training and testing resolution are the same (224 px), we observe Axial RoPE has its best performance at $k_{rope} = 8$ and is roughly the same for other values of $k_{rope}$. We observe similar effects for resolutions relatively closer to training resolution, specifically 192 and 256 px. Moving to Table 5, we report the numbers in the case where all heads have different rotation angles. In most cases, we observe Axial RoPE to outperform 2D RoPE. We speculate that this is due to the nature of the angle generator function, and we delve into its specifics in Appendix C. Both of these ablations suggest that *only a fraction of hidden dimensions are enough to impart positional information using RoPE.* In both shared and non-shared angles, we observe that Axial RoPE with a high value of $k_{rope}$ is superior to 2D RoPE for most inference resolutions, including the training resolution itself.

## 5 CONCLUSION

In this work, we propose using Rotary Position Embeddings (RoPE) instead of Relative Positional Biases (RPB), in order to achieve better performance and better accuracy. We presented empirical evidence and analysis to support this proposition. Further, to accelerate RoPE compared to RPB, we developed `fast-rope`, a fast, CUDA-based implementation of RoPE, and showed its speedup through careful model-level benchmarking. We conducted empirical analysis on two RoPE methods: Axial RoPE and 2D RoPE. We introduced a new hyperparameter, $k_{rope}$, to control the fraction of hidden dimensions used in RoPE for both implementations, and observed that applying RoPE to only half, 1/4th, or even 1/8th of the hidden dimensions is enough to reasonably introduce positional information and achieve competitive accuracy. As a result of these findings, we foresee widespread adoption of RoPE in isotropic and hierarchical vision transformers in the near future.

ETHICS STATEMENT

This work discusses methods to potential improve and speed up transformer-based vision models. While such models can be further trained to generate malicious content like deepfakes, we foresee no direct negative implications of this work.

REPRODUCIBILITY STATEMENT

We provide all relevant implementation details of experiments in this paper in Appendix E. We use open-sourced libraries for our experiments (timm, xFormers, CUTLASS) and use the official model implementations (NAT and Swin), while only making changes to accommodate RoPE. Our models can be easily trained through timm by modifying the relevant configuration files. For evaluations on ImageNet-A, R, V2 and ReaL, we use timm's validation script with relevant TensorFlow Datasets dataloaders.

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

## A  IMPLEMENTATION AND EVALUATION DETAILS FOR FAST−ROPE

### A.1  PROBLEM SIZES FOR EVALUATION

Here we note the problem size space used for benchmarking all kernels. We assume our input feature tensor to have the shape `[B, Nh, H * W, C]` and our $\theta$ tensor to have the shape `[Nh, H * W, C/4]`, where `B` represents batch size, `Nh` represents number of heads, `H, W` represent the height and width respectively and `C` is the hidden dimension. Our implementation does not expect inputs to be contiguous except in the last (i.e. channel) dimension. We list down the problem size space in Table 7. Note that our ablations are for the case where $k_{rope} = 2$. We will achieve better speedups with a higher $k_{rope}$ value.

### A.2  ADDITIONAL INFORMATION ABOUT FAST−ROPE

In Table 6, we present the average improvement in latency of performing RoPE with the generated fused kernel using `torch.compile()` and our fused implementation over eager PyTorch. The

| Prec. | Compiled over eager | | | Fused over eager | | | Fused over compiled | | |
|---|---|---|---|---|---|---|---|---|---|
| | Avg. | Min. | Max. | Avg. | Min. | Max. | Avg. | Min. | Max. |
| 16-16 | ↑ 2 % | ↓ -6 % | ↑ 975 % | ↑ 1121 % | ↑ 709 % | ↑ 1905 % | ↑ 1117 % | ↑ 33 % | ↑ 1900 % |
| 16-32 | ↑ 2 % | ↓ -5 % | ↑ 850 % | ↑ 1091 % | ↑ 727 % | ↑ 1700 % | ↑ 1088 % | ↑ 33 % | ↑ 1700 % |
| 32-32 | ↑ 1 % | ↓ -8 % | ↑ 650 % | ↑ 598 % | ↑ 333 % | ↑ 1033 % | ↑ 596 % | ↑ 33 % | ↑ 1033 % |

Table 6: **Gains using our fused kernel.** We present minimum, maximum and average gains in speed over a wide range of problem sizes. The first column signifies the precision of the feature vector and rotation angle tensor respectively. For example, `16-32` implies that the feature vector is in `float16` and the rotation angle tensor is in `float32`. Measured on A100-SXM4-80G.

full range of problem sizes is presented in Appendix A.1. Our implementation enjoys a speedup of 10 to $11\times$ in the case where the input features are in `float16`, and a speedup of almost $6\times$ when the features are in `float32`. Note that the math precision is still in `float32` in our implementation. Our implementations provide an average speedup of $9.34\times$ over `torch.compile()`.

Akin to xFormers, we do not need our inputs to be contiguous in memory, we just expect the stride of the last dimension to be 1. Our implementation supports `float16`, `bfloat16`, `float32`, `float64` data-types for the feature vector and rotation angles tensor. We use CUTLASS Thakkar et al. (2023) constructs to perform vectorized memory reads and to perform math on the accumulated arrays. For the purposes of our testing, we set the number of dimensions for RoPE to be half the dimensions in the feature vector. Intuitively, the speed benefits will increase as the fraction of dimensions decreases.

| Dimension | Possible values |
|---|---|
| B | [1, 16, 32, 64, 128] |
| Nh | [1, 3, 4, 6, 8] |
| H, W | [56, 28, 14, 7] |
| C | [32, 64, 128] |

Table 7: **Problem sizes used for evaluation.** We test over a wide range of problem sizes. We take the outer product of all these parameters to generate our problem size space. The speedup reported in the main section are average, minimum and maximum speedups of all problem sizes.

# B ADDITONAL ABLATIONS ON DESIGN DECISIONS IN ROPE VARIANTS

## B.1 ABLATION ON ANGLE GENERATOR AND POSITION CO-ORDINATES

We perform additional ablations on the angle generator function and position co-ordinates in RoPE. Since it is computationally prohibitive to experiment with all possible combinations, we choose two settings – one with shared angles and $k_{rope} = 1$ and another with non-shared angles and $k_{rope} = 2$, akin to 2D and Axial RoPE respectively. With these settings, we experiment with the two angle generators, and the two position co-ordinate systems. We present the results in Tables 8 and 9.

## B.2 ABLATION WITH THE BEST PERFORMING SETTING IN TABLES 4 AND 5

We observe that non-shared angles with $k_{rope} = 8$ performs the best across all combinations in Tables 4 and 5, on the resolution of 224 px. Motivated by this, we experiment with this configuration on all models. We present the results in Table 10

# C ANALYSIS OF ROTATION ANGLES IN 2D ROPE AND AXIAL ROPE

In Sec 3.3, we outline the differences between 2D RoPE and Axial RoPE. Through our ablations in Table 4 and 5, we eliminate the practical differences in the two variants. We now turn our attention towards the two fundamental and theoretical differences – namely the angle generators. In Figure

|  |  | Position co-ordinates | |
|---|---|---|---|
|  |  | Absolute indices | Between -1 and 1 |
| *Angle generator* | Exponential decay | 81.0 | 80.7 |
|  | Bounded log-sampling | 81.2 | 81.4 |

Table 8: Ablation on angle generator and position co-ordinates with non-shared angles and $k_{rope} = 2$.

|  |  | Position co-ordinates | |
|---|---|---|---|
|  |  | Absolute indices | Between -1 and 1 |
| *Angle generator* | Exponential decay | 81.4 | 81.0 |
|  | Bounded log-sampling | 81.0 | 80.9 |

Table 9: Ablation on angle generator and position co-ordinates with shared angles and $k_{rope} = 1$.

4, we plot the dimension-wise angles of rotation for $d = 256$ and $d = 512$. The plots give us some intuitive explanation about the disparity in multi-resolution performance of the two variants. We make the following observations:

1. Angles in Axial RoPE are higher in magnitude throughout all dimensions than 2D RoPE.

2. Angles in Axial RoPE occur in chunks, and are repeated multiple times to cover the entire feature vector.

3. Angles for 2D RoPE are monotonically decreasing in magnitude.

4. For the latter dimensions in 2D RoPE, the angles of rotation are orders of magnitude lower than for the former dimensions.

From these observations, we make two hypotheses: first, higher rotation angles imply a stronger injection of information. This explains the consistent performance of Axial RoPE, even when $k_{rope}$ is reduced to 8 or 16, but where the angles of rotation are still large in magnitude. Second, in the cases where testing resolution is higher than training resolution, we observe that Axial RoPE is roughly equal or surpasses 2D RoPE. We attribute this to the position co-ordinates assigned in Axial RoPE. Specifically, we speculate that interpolating indices to be between $(-1, 1)$, coupled with the repeating, high-magnitude angles of rotation results in a better encoding of positions in the tokens for higher test-time resolutions.

## D  NAT-S AND DINAT-S WITH AXIAL RoPE

Here, we report the accuracies for NAT-s and DiNAT-s in Table 11. We observe that both these model families enjoy the same accuracy and throughput improvements as the other models.

## E  OTHER IMPLEMENTATION DETAILS

In this section, we will report the implementation details for all models trained in the paper.

We use the official NAT[2] and Swin[3] implementations, with the only changes being to the attention module to accommodate RoPE. For the Mini, Tiny and Small variants, we follow the hyperparameters in Table 12. For the Base variants, we follow the hyperparameters in Table 13. One can easily plug these values into the corresponding fields in `timm`'s .yml files and obtain the same results. We use `timm` [4] training and evaluation scripts. We use a single node with 8 A100-SXM4 GPUs for our experiments.

---

[2]NAT
[3]Swin
[4]`timm`

|              | Original | Shared PE, $k_{rope} = 8$ |
|--------------|----------|----------------------------|
| **NAT-small**   | 83.8     | 83.8                       |
| **DiNAT-small** | 83.9     | 83.9                       |
| **Swin-small**  | 83.1     | 82.8                       |
| **ViT-small**   | 81.4     | 81.5                       |

Table 10: Ablation with shared PE and $k_{rope} = 8$.

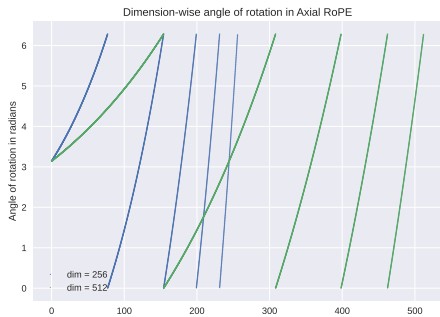
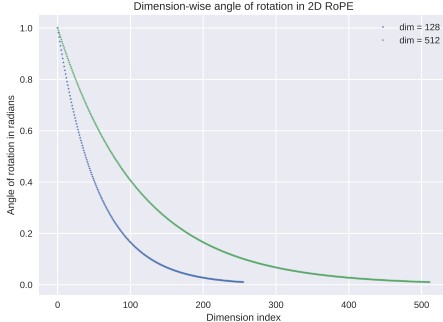

Figure 4: **Comparing rotation angles for Axial and 2D RoPE.** We illustrate the dimension-wise rotation angles for Axial and 2D RoPE for $d = 256$ and $d = 512$.

## F    ADDITIONAL EVALUATIONS ON IMAGENET VARIANTS

We have included additional evaluations on ImageNet-A, ImageNet-R, ImageNet-V2 and ImageNet-ReaL in this section.

| Model | | RPB | | | No bias | | | Axial RoPE | | |
|---|---|---|---|---|---|---|---|---|---|---|
| | | Accuracy (%) | Thru. (imgs/sec.) | | Accuracy (%) | Thru. (imgs/sec.) | | Accuracy (%) | Thru. (imgs/sec.) | |
| | | | GEMM | Fused | | GEMM | Fused | | GEMM | Fused |
| **NAT-s** | Tiny | 81.7 | 2687 | 3773 | 80.6 (-0.9) | 2884 (+7.3%) | 3894 (+3.2%) | 82.1 (+0.4) | 2779 (+3.4%) | 3759 (-0.4%) |
| | Small | 83.3 | 1700 | 2430 | 82.0 (-1.3) | 1830 (+7.6%) | 2516 (+3.5%) | 83.4 (+0.1) | 1764 (+3.8%) | 2429 (0.0%) |
| | Base | 83.6 | 1295 | 1851 | 82.7 (-0.9) | 1395 (+7.7%) | 1914 (+3.4%) | 83.7 (+0.1) | 1343 (+3.7%) | 1861 (0.5%) |
| **DiNAT-s** | Tiny | 81.8 | 2553 | 3992 | 80.7 (-1.1) | 2801 (+9.7%) | 4088 (+2.4%) | 81.9 (+0.1) | 2705 (+6.0%) | 3941 (-1.3%) |
| | Small | 83.4 | 1611 | 2572 | 82.8 (-0.6) | 1774 (+10.1%) | 2641 (+2.7%) | 83.7 (+0.3) | 1711 (+6.2%) | 2552 (-0.8%) |
| | Base | 83.8 | 1226 | 1962 | 83.0 (-0.8) | 1350 (+10.1%) | 2010 (+2.4%) | 84.0 (+0.2) | 1303 (+6.3%) | 1951 (-0.6%) |

Table 11: **ImageNet-1k classification top-1 accuracy with inference throughput.** We present the top-1 accuracy of NAT-s and DiNAT-s model families on the ImageNet-1k validation split. We observe a accuracy boost and comparable throughput with respect to RPB across both model families.

| Hyperparameter | Value |
|---|---|
| Total epochs | 310 |
| Warmup epochs | 20 |
| Cooldown epochs | 10 |
| Per GPU batch size | 128 |
| Warmup LR | $1e-6$ |
| Minimum LR | $5e-6$ |
| Base LR | $1e-3$ |
| LR schedule | Cosine annealing with linear warmup |
| Weight decay | $5e-2$ |
| EMA | False |

Table 12: Hyperparameters for Mini, Tiny and Small variants.

| Hyperparameter | Value |
|---|---|
| Total epochs | 310 |
| Warmup epochs | 50 |
| Cooldown epochs | 10 |
| Per GPU batch size | 128 |
| Warmup LR | $1e-6$ |
| Minimum LR | $5e-6$ |
| Base LR | $1e-3$ |
| LR schedule | Cosine annealing with warmup |
| Weight decay | $5e-2$ |
| EMA | True |

Table 13: Hyperparameters for Base variant.

| Model | | ImageNet ReaL | | | ImageNet V2 | | |
|---|---|---|---|---|---|---|---|
| | | *RPB* | *No bias* | *Axial RoPE* | *RPB* | *No bias* | *Axial RoPE* |
| NAT | Mini | 87.20 | 86.90 | **87.51** | 70.82 | 70.31 | **71.55** |
| | Tiny | 87.70 | 87.41 | **87.96** | 72.00 | 72.24 | **73.27** |
| | Small | 88.03 | 87.91 | **88.14** | 73.23 | 72.62 | **73.76** |
| | Base | 88.58 | 88.34 | **88.70** | 74.11 | 73.72 | **74.34** |
| DiNAT | Mini | 87.02 | 86.74 | **87.22** | **71.23** | 70.47 | 71.07 |
| | Tiny | 87.51 | 87.45 | **87.66** | 71.95 | 71.65 | **72.43** |
| | Small | 87.91 | 87.74 | **88.17** | 73.57 | 72.95 | **73.70** |
| | Base | 88.58 | 88.24 | **88.55** | 74.05 | 73.65 | **74.51** |
| Swin | Tiny | 86.55 | 85.95 | **86.89** | 69.40 | 68.45 | **70.29** |
| | Small | 87.64 | 86.86 | **87.77** | 71.93 | 70.68 | **72.52** |
| | Base | 87.94 | 87.32 | **88.02** | 72.52 | 71.89 | **72.98** |
| ViT | Small | 86.64 | 84.91 | **86.77** | 70.25 | 67.56 | **70.64** |
| | Base | 86.94 | 85.77 | **87.15** | 70.89 | 69.51 | **71.33** |

Table 14: Top-1 accuracies on ImageNet ReaLBeyer et al. (2020) and ImageNet V2 Recht et al. (2019).

| Model | | ImageNet A | | | ImageNet R | | |
|---|---|---|---|---|---|---|---|
| | | *RPB* | *No bias* | *Axial RoPE* | *RPB* | *No bias* | *Axial RoPE* |
| NAT | Mini | 12.77 | 10.89 | **13.27** | 29.60 | 28.71 | **31.36** |
| | Tiny | 17.12 | 14.95 | **18.81** | 31.59 | 30.65 | **32.35** |
| | Small | 19.27 | 18.07 | **20.51** | 33.41 | 32.36 | **33.73** |
| | Base | 22.03 | 20.35 | **23.68** | 35.11 | 34.52 | **35.76** |
| DiNAT | Mini | 13.20 | 11.32 | **13.36** | 30.77 | 29.21 | **30.81** |
| | Tiny | 16.12 | 16.24 | **17.48** | 31.40 | 30.94 | **32.26** |
| | Small | 20.35 | 19.97 | **22.04** | 32.97 | 32.26 | **34.16** |
| | Base | 22.41 | 21.35 | **23.93** | 36.04 | 34.48 | **36.96** |
| Swin | Tiny | **10.03** | 7.73 | 10.00 | 27.29 | 24.58 | **28.06** |
| | Small | 16.43 | 12.57 | **17.09** | 31.13 | 27.80 | **32.25** |
| | Base | 18.77 | 15.81 | **19.32** | 31.96 | 29.20 | **32.84** |
| ViT | Small | 11.27 | 7.93 | **11.99** | 28.87 | 22.24 | **31.24** |
| | Base | **14.00** | 9.96 | 13.92 | 31.70 | 24.87 | **33.40** |

Table 15: Top-1 accuracies on ImageNet A Djolonga et al. (2021) and ImageNet RHendrycks et al. (2021).

