# OpenReview forum: "Revisiting Positional Information in Transformers in the era of Fused Attention"
_ICLR.cc/2025/Conference — Submitted to ICLR 2025_

### Official Review · Reviewer_c8uQ · 2024-11-02

**Soundness:** 4
**Presentation:** 4
**Contribution:** 2
**Rating:** 6
**Confidence:** 2

**Summary:**

The paper proposes Rotary Postional Embeddings as a replacement for relative positional bias in transformers. They shows that it is leads to better accuracy and faster implementation. The paper tries to tackle the issue of latency when RPB is used with modern attention implementations such as flash attention.

**Strengths:**

Instead of applying RPB which slows down the attention computation in flash attention modules, the paper devices a way to add positional embeddings before attention is computed.
Since RoPE implementation is not dependent on which attention module is used, it can be integrated with any of the modern fast modern attention implementation unlike RPB.
Developed an efficient CUDA implementation for RoPE with an easy-to-use Python wrapper.
Found that applying RoPE to a fraction of embedding is enough.

**Weaknesses:**

The fact that RoPE will improve performance in ViT is not a novel idea and has already been shown in
P. Jeevan and A. Sethi, "Resource-efficient Hybrid X-formers for Vision," 2022 IEEE/CVF Winter Conference on Applications of Computer Vision (WACV), Waikoloa, HI, USA, 2022, pp. 3555-3563, doi: 10.1109/WACV51458.2022.00361. This paper has not been cited.
The scope of this paper is limited to cases when fused attention kernel is used. When RPB is introduced in this case, it hampers the fast attention compute.
Most of the paper is just a review of postional embeddings and biases.
Table 3 shows best performance when there is no bias introduced. The paper does not explain why this is so and also why even RoPE is needed then.
The ablations and experiments needs to more elaborated.

**Questions:**

Explain in detail the actual contributions of this paper and what novel ideas where brought in?
Is Axial RoPE your contribution or has it been taken from another paper and you just did a lot of analysis on it?
Why do we even need RoPE if no bias gives best results?

---

> ### Author Response · Authors · 2024-11-16
>
> We thank the reviewer for their insightful comments.
>
> _**Regarding novelty**_, we request the reviewer to take a look at the common response for clarification.
>
> _**Regarding applicability outside fused attention**_, we would like to clarify that RoPE and our implementations of RoPE can be used with both fused or unfused attention alike. In fact, we observe significant throughput gains in both cases, as explained in Table 3.
>
> _**Regarding Table 3,**_ Table 3 shows throughput of the respective models with RPB, Axial RoPE and without any bias. The “No bias” case implies there is no additional computation done in the model to impart positional bias. Thus, it is expected that the models without any bias would be slightly faster, since they don’t perform a chunk of computation as compared to other models with biases.
>
> **We will cite the mentioned work in the Related Works section. Thanks for bringing the same to our attention!**

---

### Official Review · Reviewer_Ue2F · 2024-11-04

**Soundness:** 2
**Presentation:** 3
**Contribution:** 2
**Rating:** 5
**Confidence:** 4

**Summary:**

This paper explores the use of Rotary Position Embeddings (RoPE) in vision transformer models and provides an analysis of the differences between Relative Positional Biases (RPE) and RoPE through empirical experiments. Additionally, it proposes a fused CUDA-based implementation to improve inference speed. The paper also presents various design choices for RoPE, supported by experimental results.

**Strengths:**

- The paper is well-written and easy to follow. The figures demonstrate the ideas clearly.

**Weaknesses:**

- **Unclear Contribution:** The novelty of the paper is uncertain, as RoPE has been previously applied to vision transformers. Notably, Heo et al. (2024) and Chu et al. (2024) have already explored RoPE in this context, with 2D RoPE in particular resembling Heo et al.'s work. Further discussion is needed to clarify the differences between the current approach and previous implementations. A comparison in the experimental section is also recommended. Additionally, the authors should consider reorganizing the contribution sections, as the first two contributions appear unconvincing.

- **Inconclusive Results:** The paper lacks a definitive conclusion regarding the performance of 2D RoPE versus Axial RoPE. For instance, Table 4 shows that 2D RoPE outperforms Axial RoPE, warranting further discussion.
- **Limited Generalization Testing:** The paper does not assess the generalization ability of Axial RoPE across downstream tasks (e.g., detection and segmentation). Additional experiments to showcase RoPE’s generalization potential are recommended.

**Questions:**

- In comparing Table 4 and Table 5, the shared angles consistently outperform the non-shared angles. Why, then, did the authors choose to use non-shared angles in Axial RoPE?

---

> ### Author Response · Authors · 2024-11-16
>
> We thank the reviewer for their comments and constructive criticism. We would like to clarify the reviewer’s concerns here:
>
> _**Regarding unclear contribution**_, we request the reviewer to look at the common response for clarification.
>
> _**Regarding inconclusive results, shared angles and limited generalization**_:
>
> **Following your suggestion, we are currently performing the experiments with k=8 and shared angles across all heads. We will share an update when we have some numbers regarding the same.** At the time of submission, we chose to stick to a reasonable default, since performing experiments with all possible settings is infeasible. We could not include experiments on detection and segmentation owing to limited academic compute.

---

> ### Author Response · Authors · 2024-11-23
>
> **Following your suggestion, we ran experiments with shared angles for all heads and k=8 for the `small` variant of all models.**
>
> We observed that while NAT and DiNAT do not exhibit any noticeable changes, we observe that Swin-small degrades by 0.3%, while ViT-small improving by 0.1%. From this set of experiments, we conclude that the actual choice of hyperparameters would depend on the model at hand the attention mechanism used in the same. From the above results, we would only hypothesize that since NAT and DiNAT already have an implicit spatial bias built into the attention mechanism, they are not significantly affected by changes in the implementation of RoPE. However, since Swin and ViT do not have any implicit spatial bias, they are more sensitive to changes in RoPE's implementation.
>
> _Thanks a lot for your suggestion!_
>
> Results from additional ablations:
> |                 | Original | Shared PE, $k_{rope}=8$ |
> |-----------------|:----------:|:-------------------------:|
> | **NAT-small**   |   83.8   |           83.8          |
> | **DiNAT-small** |   83.9   |           83.9          |
> | **Swin-small**  |   83.1   |           82.8          |
> | **ViT-small**   |   81.4   |           81.5          |
>
> _**Table 1:**_ Ablations for `small` variant of all models with shared PE across all heads and $k_{rope}=8$.

---

> > ### Comment · Reviewer_Ue2F · 2024-11-24
> >
> > Thank you for the authors’ response. After thoroughly reviewing both the common response and the specific reviews, I still have some concerns regarding the experimental conclusions and whether the claimed contributions meet the standards of ICLR. The contributions highlighted in the common response appear incremental. Additionally, the inconclusive results comparing 2D RoPE and Axial RoPE remain insufficiently addressed, and the generalization ability of RoPE has not been evaluated. To make the paper stronger, the contributions need to be clearly clarified, and include more significant technical or conceptual novelty. Overall, I believe the current paper requires some revisions to be accepted.

---

### Official Review · Reviewer_uvBo · 2024-11-05

**Soundness:** 3
**Presentation:** 3
**Contribution:** 3
**Rating:** 8
**Confidence:** 2

**Summary:**

The paper proposes using RoPE embeddings, a popular and widely used method for LLMs for vision transformers motivated by imperial gains in accuracy and efficiency when applying to multiple models of various sizes. For this, they extend RoPE to fit image space and tackle the challenge of implementing it and studying multiple rotary positional embedding implementations.

**Strengths:**

- The expansion of RoPE to images is presented clearly and makes intuitive sense
- The paper does a great job of motivating and describing the CUDA implementation.
- It is deeply appreciated that they go deeper and explore multiple different Rotary Positional Embeddings and report the comparisons

**Weaknesses:**

- While a lot of small improvements are introduced in the method (3.3.2) the support or estimation of impact is somewhat lacking in the results.
- While the impact of k is detailed and appreciated, the measurement of performance is limited to accuracy and makes it hard to understand the gains or sacrifices associated with the implementations.
- The paper claims "noteworthy gains" across models, however the gains in Table 2 seem relatively limited (0.1-0.2) in most cases.
- Limited novelty, while the expansion of RoPE makes sense, the novelty both in terms of method and results might be limited.

**Questions:**

- Could you expand on the justification for RoPE's superior performance compared to RPB, beyond the intuitive explanations provided?
- Would the gains in efficiency scale to larger model size and resolution combinations?

---

> ### Author Response · Authors · 2024-11-16
>
> We thank the reviewer for their comments and constructive criticism. We would like to clarify the reviewer’s concerns here:
>
> _***Regarding weakness (1)***_, we want to clarify Section 3.3.2 is a review of different rotary embedding approaches [1,2]. Our ablations have been limited to k_rope and shared angles in Tables 4 and 5, but **following your suggestion, we will include ablations of all design choices mentioned in Section 3.3.2**. Thank you for bringing this to our attention!
>
> _**Regarding weakness (2)**_, since we were limited to image classification due to compute constraints, we report ImageNet-1k accuracy and inference throughput for different positional biases. It is indeed commonly accepted that RoPE has equal or better accuracy than RPB, and thus we focus our efforts on showing why RoPE is more scalable than RPB.
>
> _**Regarding weakness (3)**_, we would like to point out that even though the gains seem marginal, they cannot be achieved trivially. The gains also support the hypothesis that RoPE consistently outperforms RPB, while being hardware-friendly at the same time.
>
> _**Regarding weakness (4)**_,  we request the reviewer to look at the common response for clarification.
>
> _**Regarding question (1)**_, we hypothesize that RoPE outperforms RPB (in terms of accuracy) because it simply provides the model a spatial bias, without having the need to learn the same. In contrast to this, RPB requires the model to learn this spatial bias itself, which might not be optimal given the data-hungry nature of transformer-based vision models.
>
> _**Regarding question (2)**_ yes, we expect the gains to effectively scale to larger models and resolutions. As the model size and resolution increases, RPB will indeed become a larger bottleneck due to the nature of fused attention mechanisms. In addition, RoPE will continue to scale up training efficiently as well, since it is a simple elementwise operation in both the forward and backward pass. Attention biases become a much larger bottleneck in training, since their backward pass is a reduction operation, and typically bound by memory bandwidth. We can see this in comparative training throughputs of RoPE and RPB models. For example, a ViT Base model with RPB has a training throughput of 6071 images/sec, whereas a ViT Base with Axial RoPE has a training throughput of 6596 images/sec, thereby observing a training speedup of ~8.5%.
>
> **References:**
>
> [1] Crowson, Katherine, et al. "Scalable high-resolution pixel-space image synthesis with hourglass diffusion transformers." Forty-first International Conference on Machine Learning. 2024.
>
> [2] Heo, Byeongho, et al. "Rotary position embedding for vision transformer." arXiv preprint arXiv:2403.13298 (2024).

---

> ### Author Response · Authors · 2024-11-23
>
> **Following your suggestion, we perform ablations of all design choices in Table 1** -- position co-ordinates and angle generator. We perform the ablations while keeping the settings on Axial and 2D RoPE for the other two design choices (k and sharing of angles across heads). We will add the same in the manuscript.
>
> _Thanks a lot for the suggestion!_
>
> Results from additional ablations:
>
> |                 |                      | Position co-ordinates |                  |
> |:---------------:|:--------------------:|:---------------------:|:----------------:|
> |                 |                      |    Absolute indices   | Between -1 and 1 |
> | **Angle generator** |   Exponential decay  |          81.0         |       80.7       |
> |                 | Bounded log-sampling |          81.2         |       81.4       |
>
> _**Table 1:**_ Ablation on angle generator and position co-ordinates with non-shared angles and $k_{rope}=2$.
>
> |                 |                      | Position co-ordinates |                  |
> |:---------------:|:--------------------:|:---------------------:|:----------------:|
> |                 |                      |    Absolute indices   | Between -1 and 1 |
> | **Angle generator** |   Exponential decay  |          81.4         |       81.0       |
> |                 | Bounded log-sampling |          81.0         |       80.9       |
>
> _**Table 1:**_ Ablation on angle generator and position co-ordinates with shared angles and $k_{rope}=1$.

---

> > ### Comment · Reviewer_uvBo · 2024-11-26
> >
> > I thank the authors for their general and individual replies to my questions, especially for extending the ablation of the design choices. I still have concerns about the relative marginality of the improvements, however, I believe that the overall analysis makes for a good contribution. I maintain my rating.

---

### Author Response · Authors · 2024-11-16
**Common response to all reviewers**

We thank all reviewers for their insightful comments and constructive criticism.

We would like to clarify the contributions and intended message of our work here. Our work has the following objectives:

1. To motivate the community to prefer positional embeddings (RoPE or otherwise) over attention weight biases in both large and small transformer-based vision models. We also want to provide concrete insights about the usage of positional embeddings and the downsides of using attention weight biases (like Relative Positional Biases, RPB) with fused attention implementations.
2. Consequently, we intend to study two flavors of RoPE, compare them, and introduce an efficient CUDA-based implementation to expedite this shift. Previous works which use RoPE in transformer-based vision models overlook training speed and we intend to improve the case for RoPE in that direction.
3. We observe that applying RoPE to a fraction of feature dimensions per head is enough to impart positional information, and we provide empirical evidence of the same.

---

### Meta-Review · Area_Chair_gErD · 2024-12-11

**Metareview:**

This paper explores the application of Rotary Positional Embeddings (RoPE) in vision transformers by extending it to 2D and Axial RoPE and developing efficient CUDA implementation. Ablation studies analyze the design choices of RoPE, including positional coordinates, angle sharing, and the fraction of embeddings used. However, the novelty of this work is somewhat limited and the Axial RoPE design appears to be incremental. The performance gains reported are relatively minor and might not justify the added complexity in some cases. While the CUDA implementation adds value, the contributions do not meet the standards of significant technical or conceptual innovation expected for acceptance at ICLR.

**Additional Comments On Reviewer Discussion:**

- Reviewers Ue2F and c8uQ raised concerns about the novelty of the paper, particularly when compared to prior works. In evaluating the submission, I did not factor in a comparison with the concurrent ECCV'24 work in the final decision.
- Several reviewers highlighted the absence of generalization experiments beyond ImageNet-1k classification. While Reviewer uvBo acknowledged the computational advantages of RoPE, they also expressed concerns about the relatively minor performance gains. The authors' additional ablation studies demonstrated slight variations in accuracy across different configurations but failed to fully address the concerns regarding RoPE's limited impact on overall performance.

While the authors addressed some concerns, the rebuttals did not significantly strengthen the contributions. The novelty remains limited, and the lack of downstream experiments and insufficient comparisons weaken the paper's case for acceptance.

---

### Decision · Program_Chairs · 2025-01-22

Reject